psychology

visual attention, inattentional blindness, time course of attention

**Author for correspondence:**
Katherine Wood
e-mail: kmwood2@illinois.edu

# Now or never: noticing occurs early in sustained inattentional blindness

Katherine Wood and Daniel J. Simons

Department of Psychology, University of Illinois, 603 E. Daniel Street, Champaign, IL 61820, USA

(iD) KW, 0000-0002-3877-9625

People can show sustained inattentional blindness for unexpected objects visible for seconds or even minutes. Would such objects eventually be noticed given enough time, with the likelihood of noticing accumulating while the unexpected object is visible? Or, is there a narrow window around onset or offset when an object is most likely to be detected, with the chances of noticing dropping outside of that window? Across three experiments (total $n$'s = 283, 756, 488) exploring the temporal dynamics of noticing in sustained inattentional blindness, subjects who noticed the unexpected object did so soon after it onset. Doubling or even tripling the time when the unexpected object was visible barely affected the likelihood of noticing it and had no impact on how accurately subjects reported its features. When people notice an unexpected object in these sustained inattentional blindness tasks, they do so soon after the unexpected object onsets.

# 1. Introduction

As anyone who has ever tried to get the attention of a distracted friend knows, people can be remarkably oblivious to new or unexpected events when they are sufficiently engrossed in something else. Someone can be *inattentionally blind* to your waving (and inattentionally deaf to your calling out). A large body of work has attempted to delimit when people will experience inattentional blindness [1], what aspects of the unexpected event affect noticing, and how engagement in a primary task matters. Fewer studies have examined an equally interesting question: *When* do people notice? Is your friend guaranteed to notice you eventually if you keep waving? When during your efforts will they be most likely to spot you?

Most studies of the temporal characteristics of selective attention have focused on tasks in which subjects know they will have to attend to some areas or objects and ignore others; they can evaluate and establish attentional filters for all relevant aspects of the display and use those filters repeatedly across many trials.

In these circumstances, attention can be deployed to a particular location or feature in a few hundred milliseconds [2]. For example, a stimulus in a search or RSVP task can be processed in something on the order of 50 ms [3] and inhibition of distractor features can begin as early as 100 ms after stimulus onset [4].

These and other methods of studying the temporal characteristics of attention (e.g. event-related potentials) cannot be employed to study noticing in inattentional blindness due to the one-shot nature of the phenomenon. Once subjects become aware that something unexpected may appear, it is no longer unexpected.[1] Because it is essential that subjects cannot prepare for the unexpected stimulus, inattentional blindness tasks typically use one trial per subject which precludes the within-subjects approaches of other tasks.

Across different types of tasks, subjects can miss unexpected objects across a variety of time scales. A substantial proportion of subjects—25–75%, depending on the precise nature of the experiment—miss a small square flashed for 200 ms in a static display [1]. About half of subjects also miss a cross drifting across the screen for 5 s in a multiple object tracking task [6] or a gorilla striding through a basketball game for 9 s [7]. However, few studies have systematically examined the effect of exposure time on noticing within a single task. When people do notice an unexpected object, at what point during the task does it reach awareness? Would longer exposure to the same object lead to more noticing?

One of the original studies of selective looking, in which participants viewed the same display for different lengths of time, offers a potential answer to these questions [8]. Subjects viewed a video of two basketball teams, with instructions to track the passes made by one team while ignoring those made by the other team. During the action, a woman holding an umbrella walked through the display. In one condition, the woman passed all the way across the screen and was visible for about 5.5 s. In another condition, the experimenters stopped the video when she was about halfway across, amounting to roughly 2 s of exposure. The primary purpose of the study was not to examine the effect of exposure on noticing, but it did appear to make a difference: 34% of subjects in the 5.5 s condition noticed the umbrella woman, whereas just 7% did in the 2 s condition. More exposure to the unexpected object apparently increased the probability that it would be noticed. As the study itself noted, however, there may have been critical content differences between the first 2 s of the video and the remaining 3.5 s (other than just exposure time) that contributed to differences in noticing.

More recent studies examined the influence of exposure time on noticing with a multiple object tracking task in which subjects monitor a subset of objects and count the number of times they bounce off the edge of the display [9]. One experiment found little difference in noticing for a 'fast' unexpected object that took 5 s to cross the display and a 'slow' one that crossed in 9 s. However, the study used relatively small samples ($n = 25$ per condition) with low rates of noticing for the first appearance of the unexpected object (4 and 5 noticers in the slow and fast conditions, respectively), so the study does not permit definitive conclusions about whether or not exposure time mattered for noticing.

Another recent study also varied exposure time to the unexpected object by varying its speed in a multiple object tracking task [10] and observed higher noticing rates for the slower object that was on screen for longer. Noticing rates were comparable when the fast and slow objects were on-screen for the same amount of time, suggesting that increased exposure to the objects, rather than the speed difference, increased the likelihood of noticing.

A much earlier and more unusual study, intended as a test of how people perceive and respond to seemingly paranormal events, provided early evidence of inattentional blindness for real-world events and also collected data on approximately when observers noticed the unexpected event [11]. The experimenter dressed in a sheet and walked back and forth across the stage of a movie theatre while a trailer was playing before the film. The 'ghost' was visible for 50 s, and 32% of the theatre audience did not report seeing it. Of the 68% who did notice the ghost, just over half saw it in the first 5 s it was visible (inferred from the part of the ghost's walk that they reported).

The methods used in each of these studies provide different information about the time course of noticing. Varying the amount of time the unexpected object is visible tests how much noticing increases with additional exposure time. Asking when subjects first noticed the object narrows down when noticing occurs, particularly if noticing does not vary between exposure times. If the same proportion of subjects notice the unexpected object with a long and short exposure, the location reports can clarify why (e.g. noticing occurs at offset). In order to get a more complete picture of the time course of noticing, we can collect both kinds of data in the same experiment by varying the

---

[1]While repeated inattentional blindness in a single task can occur, the proportion of non-noticing subjects drops substantially with each repeated presentation of the 'unexpected' object [5].

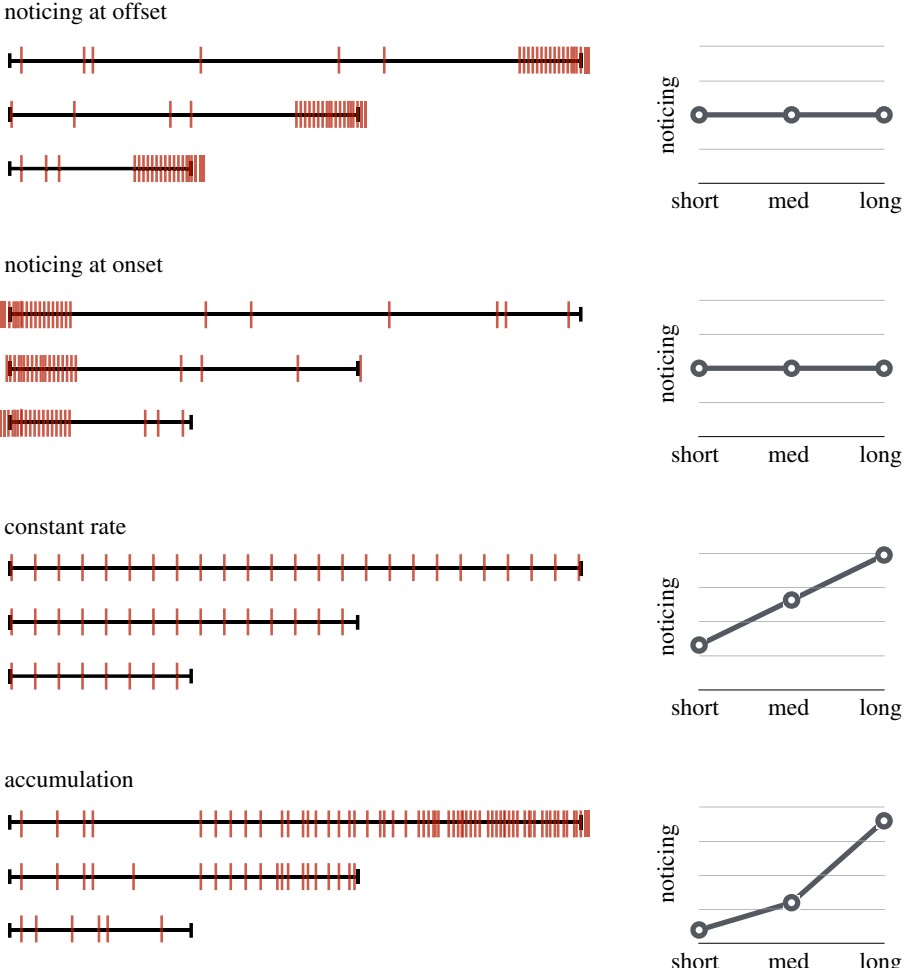

**Figure 1.** Example models for noticing events. The timelines on the right are sample distributions of location reports, with onset on the left and offset on the right. The plots on the left show the corresponding shape of the noticing rates across time intervals. Some models, such as noticing at onset/offset, predict a constant rate of noticing across time, but different patterns of localization. Others predict a change in noticing across time intervals.

amount of time the unexpected object is on-screen, keeping all other aspects of the display identical, and asking subjects to report where the object was when they first noticed it. Subjects are fairly accurate at localizing an unexpected object when they do notice it [12], and the location reports should be sufficient to disambiguate cases where the noticing rate does not vary with exposure time.

If noticing is triggered by a transient event such as the onset or offset of the unexpected object, then noticing rates should be constant across exposure times, and the location reports should cluster around the location of that onset or offset (figure 1). If noticing is a stochastic process [13], greater exposure time should provide more opportunity to notice, so noticing rates across participants should be higher with increasing exposure time [8,10]. That pattern would match the intuitive idea that if your friend does not see you at first, it will help to keep waving.

# 2. General methods

## 2.1. Subjects

The University of Illinois Institutional Review Board (IRB) waived the requirement for signed consent due to the low-risk nature of the experiment. Prior to accepting the HIT ('Human Intelligence Task', the term for the jobs posted to Amazon's Mechanical Turk service), subjects were shown an information screen that provided experimenter and IRB contact information. It explained that their responses would be anonymous, described how their data would be used, and noted that their

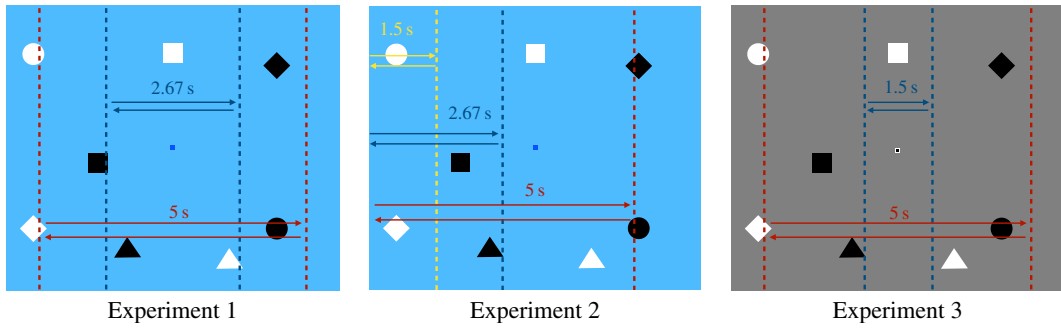

**Figure 2.** The paths taken by the unexpected object in each experiment. The unexpected object always travelled centred in the display. The dotted lines indicate the edges of the invisible occluders that the unexpected object emerged from and disappeared behind. Note that in Experiment 2, the depicted trajectories could also be positioned at the right edge of the display.

participation was voluntary. All subjects were US-based workers recruited through Amazon's Mechanical Turk (MTurk) service who had completed at least 100 HITs and had a HIT approval rating of at least 95%. We used TurkGate [14] to exclude subjects who had previously participated in experiments from our laboratory based on their worker ID. Subjects were directed to an external website running the experiment in Javascript, and upon finishing the experiment, they received a completion code which they entered on MTurk to receive payment ($0.30).

Subjects were automatically recruited in batches of up to nine using the boto3 Mechanical Turk SDK (https://github.com/boto/boto3). When we passed the recruitment threshold for an experiment, recruitment stopped and no further HITs were posted. Based on previous experiments using a similar procedure and small-sample pilot studies for this project, we anticipated a 50% exclusion rate.

## 2.2. Materials and procedure

All experiment methods, analysis plans and code were preregistered on the Open Science Framework (OSF) [15]. Each experiment was preregistered separately, prior to data collection for that experiment. Each experiment's project page includes anonymized data, all experimental materials and code, preregistration documentation, and demo versions of the task. Pilot data are also available on OSF, where applicable.

Upon accepting the HIT and navigating to the external website, subjects viewed an instruction screen explaining that they would see a group of eight objects, four white and four black. Subjects were randomly assigned to attend white or black, and were told to count how many times the 'attended' colour of object bounced off the edges of the display area (ignoring the bounces of the other objects). They were advised to fixate on a central fixation square throughout the task.

After reading the instructions, subjects proceeded to three trials of counting bounces. The display window was $600 \times 700$ pixels. It was light blue (#58ACFA) in Experiments 1 and 2 and mid-grey (#808080) in Experiment 3 (figure 2). A $10 \times 10$ pixel fixation square was centred in the display, dark blue (#0000FF) in the first two experiments and black with a white border in the third. The display contained four black objects and four white objects. Each set featured a square (40 by 40 pixels), a triangle (50 pixel base, 50 pixel height), a diamond (56 pixel width by 56 pixel height) and a circle (46 pixel diameter). The objects were randomly placed in the display area at the start of each trial and remained stationary for 1 s before they began moving to give subjects time to prepare for the start of the trial.

The objects moved between 66 and 198 pixels per second and randomly increased or decreased their velocity by 66 pixels per second after a randomly selected period ranging from 300 to 1000 ms, with the constraint that objects never moved faster than 198 pixels per second or slower than 66 pixels per second. Objects occluded one another when they crossed paths, but always passed behind the fixation square. When objects came into contact with the edge of the display, they 'bounced' off at a 45° angle from the edge at the same speed, with a reversed horizontal direction if they contacted the left or right edge of the display and a reversed vertical direction if they contacted the top or bottom edge. After 15 s of motion, the trial ended and subjects were prompted to enter their count of the bounces into a text box which only permitted integer responses.

On the third trial, an additional unexpected object passed through the display. Although some parameters varied by experiment (see each experiment's method for details), the object was always a cross ($40 \times 40$ pixels with arms 14 pixels thick) and travelled at 132 pixels per second, approximately

**Table 1.** The number of subjects excluded under each rule in each experiment. Subjects could have been excluded under multiple rules, so the total for the experiment may not match the raw sum of all exclusions.

| exclusion rule | Experiment 1 | Experiment 2 | Experiment 3 |
|---|---|---|---|
| miscounted by more than 50% on more than two trials | 190 | 367 | 333 |
| reported being younger that 18 | 0 | 0 | 1 |
| reported needing vision correction but not wearing it | 32 | 105 | 82 |
| reported a technical problem with the experiment | 107 | 140 | 131 |
| reported prior experience with inattentional blindness | 24 | 37 | 26 |
| had a severity index greater than 1.78 on the Farnsworth D-15 | n.a. | n.a. | 182 |
| total excluded | 243 | 517 | 516 |
| total retained | 283 | 756 | 488 |
| total recruited | 526 | 1273 | 1004 |

the average velocity of the display objects. It always offset with 2 s remaining in the trial, although the amount of time it was on-screen and the location at which it onset and offset varied by experiment. It always travelled horizontally through the display at the vertical midpoint, and whether it crossed left-to-right or right-to-left was random for each subject.

Following the third trial, subjects entered their bounces as usual, but were then asked whether they had noticed anything new on the last trial that had not appeared in earlier trials. Next they were asked to indicate the new object's shape from a drop-down menu of options and its colour, either from a drop-down menu (Experiments 1 and 2) or a continuous colour slider (Experiment 3). Subjects were then presented with a two-third scale image of the display rectangle and fixation square, as well as a scaled-down version of the unexpected object as it had appeared on the critical trial. The unexpected object's icon started in the upper-left corner of the scaled-down display, and subjects were instructed to move the unexpected object to the point in the display where they first noticed it.

Subjects next reported whether they needed vision correction, defined as 'glasses or contacts', and if they were wearing it during the experiment, then indicated any technical issues they experienced. Finally, they were asked whether they had any prior experience with inattentional blindness tasks. After completing the questionnaire, they were given the completion code for the experiment. In total, the experiment took most participants approximately 3–5 min to complete.[2]

## 2.3. Analysis procedure

Data were analysed in R [17] using the packages dplyr [18], purrr [19], tidyr [20], ggplot2 [21], viridis [22], ggforce [23] and circular [24]. For all analyses, we report point estimates for values of interest with 95% bootstrapped confidence intervals calculated via the percentile method [25]. For comparisons of interest, we also calculate difference scores and their 95% bootstrapped confidence intervals. All estimates and comparisons were preregistered on OSF.

## 2.4. Exclusion criteria

Our preregistered criteria excluded data from subjects who reported being younger than 18 years old; whose bounce counts erred by more than 50% in either direction on two or more trials; who reported needing vision correction but not wearing it during the experiment; who reported that the experiment lagged, froze or had some other technical problem; or who reported prior experience with inattentional blindness tasks. In Experiment 3, we also excluded participants with a confusion index greater than 1.78 on the Farnsworth D-15 task that is designed to measure colour vision [26,27]. For a detailed breakdown of the exclusions in each experiment, see table 1.

---

[2]We do not typically include full-attention trials in online experiments. The original studies of inattentional blindness presented the unexpected object briefly and needed to verify that it was visible when people were looking for it [1]. For these dynamic tracking tasks, there is little concern about visibility; given their size, high contrast, and extended time on-screen, the unexpected objects are well above threshold for visibility. Indeed, other researchers using online experiments to study inattentional blindness do not routinely employ full-attention trials [5,16].

# 3. Experiment 1

Does increasing the exposure time to an unexpected object also increase the likelihood that it is noticed? Although this idea is intuitive and has some tentative support [8,10], no previous study has systematically compared the effect of exposure times on noticing under otherwise identical conditions. Experiment 1 compared noticing of an unexpected object that was visible for either a long exposure of 5 s or a shorter exposure of 2.67 s, corresponding to the unexpected object crossing either 80% of the total width of the display or 40% of the total width display. A 5 s exposure is typical for this sort of sustained inattentional blindness task [6], and these exposure durations are similar to those used in previous studies of the influence of exposure time on noticing [8]. If noticing is a stochastic or accumulative process, with greater time leading to more noticing, then noticing rates should be lower in the 2.67 s condition because those participants have 2.33 fewer seconds to spot the unexpected event; if noticing is instead driven by an onset or offset event, then noticing rates should not differ between the conditions.

In addition to the overall noticing rates, we also assessed where those participants who noticed the unexpected object first saw it. This information provides a more fine-grained estimate of the time course of noticing, as their location reports indicate *when* they noticed the unexpected object. If most reports fall close to offset, for example, that would indicate a tendency to notice the object late, while clustering near onset would indicate early noticing. In both cases, however, the noticing rates would be the same across exposure durations, since noticing would be triggered by an event common to all presentations (figure 1). To verify that these location reports represent a meaningful signal about when people noticed the unexpected object, we included a condition with no unexpected object in this experiment. This condition will provide the true random baseline against which to compare the localization reports from the other conditions.

## 3.1. Methods

The materials and preregistration for this experiment are available at https://osf.io/yekzc/. A demonstration of the task may be viewed at simonslab.com/mot/temporal_mot_demo.html.

### 3.1.1. Subjects

We aimed to recruit 500 subjects with the goal of collecting usable data (i.e. after exclusions) from 100 subjects in each unexpected object condition and 50 subjects with no unexpected object. We recruited according to the procedure in the General methods and ended up with 526 subjects in total.

### 3.1.2. Materials and procedure

The primary task and post-survey questionnaire were as described in the General methods.

The unexpected object in this experiment was a mid-grey (#808080) cross. Subjects were randomly assigned to one of three possible unexpected object conditions. In the short-duration condition, the unexpected object appeared on-screen for 2.67 s, onsetting 10.33 s into the trial by emerging gradually from behind an invisible occluder positioned 210 pixels from the edge of the display and travelling 280 pixels before offsetting behind another invisible occluder. In the long-duration condition, the unexpected object appeared for 5 s, onsetting 8 s into the trial from an invisible occluder 70 pixels in from the edge and travelling for 560 pixels before offsetting (figure 2). In the no-unexpected-object condition, the procedure was identical to the other two conditions except that no additional object appeared on the critical trial. Subjects had a 2 in 5 chance of being assigned to either unexpected object condition or a 1 in 5 chance of being assigned to the no-unexpected-object condition.

## 3.2. Results and discussion

Prior to analysis, we excluded 243 subjects (46% of our sample) according to the criteria outlined in the General methods, leaving 283 in the analysis ($n = 54$ in the no-unexpected-object condition, $n = 104$ in the 2.67 s exposure condition, and $n = 125$ in the 5 s exposure condition).

### 3.2.1. Noticing

As specified in our preregistration, we coded subjects as having noticed the unexpected object if they were assigned to a condition that had an unexpected object, reported noticing something new and correctly reported either the object's shape or its colour.

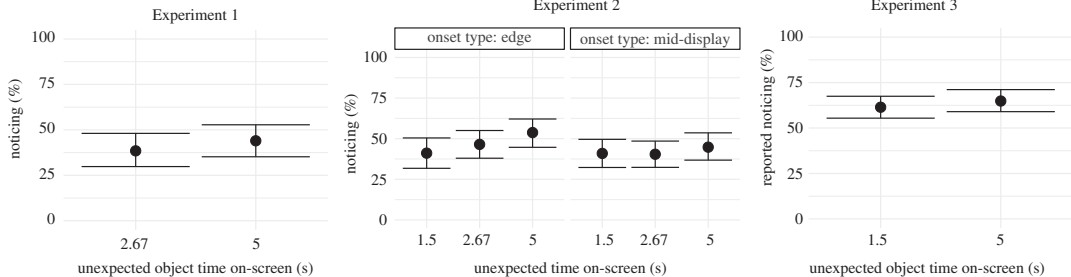

**Figure 3.** Noticing rates or self-reported noticing for subjects in each experiment, broken down by unexpected object exposure time. Error bars are 95% bootstrapped confidence intervals.

44.0% (95% CI = 35.2–52.0) of subjects in the 5 s exposure time condition noticed the unexpected object, and 38.5% (95% CI = 29.8–47.1) of subjects in the 2.67 s exposure time condition noticed it for a difference of 5.5 percentage points (95% CI = −8.1–18.0; figure 3).

### 3.2.2. Location reports

For the location analysis, we looked separately at the vertical and horizontal localization of the unexpected object. Recall that the object always travelled along a horizontal path at the vertical midpoint of the display. Although vertical localization of the unexpected object is not informative about when people noticed the unexpected object, it does indicate whether subjects are placing the object in the area it actually appeared. The pattern of localization still differed between noticers and non-noticers. Noticers fairly consistently placed the object near the horizontal midline ($M$ = 282 pixels, s.d. = 67; actual midline = 300 pixels). The average placement for non-noticers was also near the midline, but with far greater variability ($M$ = 313 pixels, s.d. = 120). The average placement for those in the no-unexpected-object condition was also near the midline, but again with greater variability than for notices ($M$ = 311, s.d. = 153).

To estimate the timepoint of noticing, the position along the horizontal midline is more critical. Recall that participants were randomly assigned to experience an unexpected object moving from right to left or from left to right. The object placement for noticers was tied to the actual onset location they experienced, with placement clustering near the onset position (figure 4). There is especially clear separation in the 5 s condition, when the left- and right-side onset points are further apart (the lower right panel for Experiment 1 in figure 4). Eighty-one per cent of noticers placed the unexpected object on the onset side of fixation. By contrast, non-noticers and subjects in the no-unexpected-object condition tended to place the object around fixation with a large spread (left column for Experiment 1 in figure 4; also see table 2). Forty-eight per cent of non-noticers and 33% of subjects in the no-unexpected-object condition placed the object on the onset side of fixation.

Some noticing subjects placed the unexpected object closer to the edge of the display than it ever appeared; these instances are most visible in the 2.67 s condition. This may reflect a tendency for some subjects to extrapolate the unexpected object's location based on its velocity; if they notice it coming from the left, for instance, they might over-correct and place it closer toward the left edge than at its actual onset point. This sort of error might reflect an actual misperception of the unexpected object's location as a result of its motion [28], noise in the precision of their localization (leading to random spread around the onset point) or a deliberate attempt to put the object where they believe it onset rather than where they actually noticed it. In general, though, subjects who noticed the unexpected object positioned it close to its actual onset location with some precision (table 2).

Collectively, Experiment 1 showed that noticing occurred fairly early after onset, and additional exposure to the unexpected object increased the chance of noticing it only slightly. These results contrast with earlier studies in which reducing the exposure also substantially decreased the proportion of subjects noticing the unexpected stimulus [8,10].

The idea that noticing occurs soon after onset and does not benefit substantially from additional exposure time is supported by the localization data. The pattern of localization reports for noticing subjects—consistent vertical placement and horizontal placement near the onset location—differed dramatically from that produced by non-noticing subjects and subjects who were not presented with an unexpected object. This difference in localization suggests that these reports are based on detection of the unexpected object and are at least coarsely reliable. Furthermore, there was no obvious

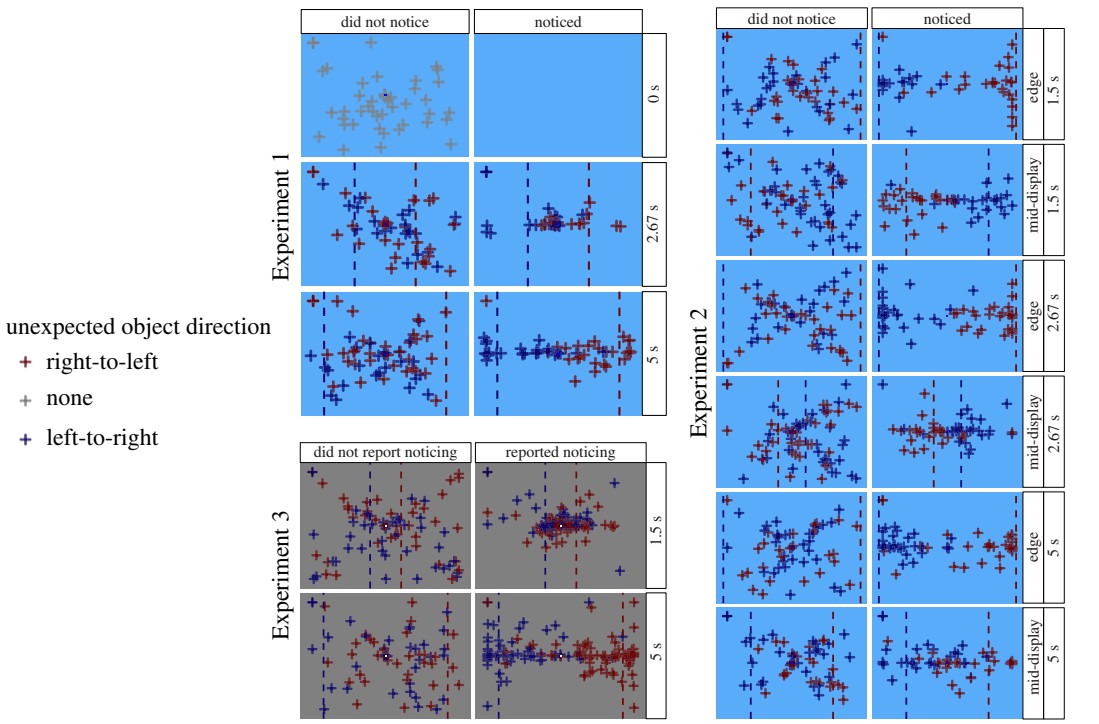

**Figure 4.** Scatterplots of the location reports from each subject in each condition. The dashed vertical lines indicate the onset points in each condition, and are colour-coded according to the direction the unexpected object travelled after it appeared. Each panel shows both the left-to-right and right-to-left variant for a particular condition, colour-coded accordingly.

**Table 2.** Average Euclidean distance, in pixels, of subjects' location reports for the unexpected object from the onset location, fixation location and offset location in Experiment 1. For each condition and direction of motion, we averaged the distance between each individual location report and the location of the onset in that condition, fixation and the offset in that condition.

| noticed unexpected object | time on-screen | unexpected velocity | mean distance of reports to fixation (s.d.) | mean distance of reports to onset (s.d.) | mean distance of reports to offset (s.d.) |
|---|---|---|---|---|---|
| no | 0 | none | 193.2 (138.8) | n.a. | n.a. |
| no | 2.67 | right to left | 151.5 (133.5) | 202.7 (111.6) | 228.6 (121.4) |
| no | 2.67 | left to right | 127.6 (111.7) | 215.4 (115.7) | 166.4 (95.9) |
| no | 5 | right to left | 165.6 (124.2) | 329.3 (156.4) | 304.9 (129.8) |
| no | 5 | left to right | 158.6 (112.7) | 288.7 (125.1) | 339.2 (136.9) |
| yes | 2.67 | right to left | 86.3 (85.7) | 127.9 (56.8) | 194.3 (107.1) |
| yes | 2.67 | left to right | 140.6 (156.1) | 155.5 (90.7) | 262.8 (159.1) |
| yes | 5 | right to left | 204.4 (116.8) | 155.7 (130.3) | 460.1 (122.3) |
| yes | 5 | left to right | 184.4 (120.9) | 156.8 (115.5) | 439.1 (151.3) |

difference between the locations reported by subjects who missed the unexpected object and those who were not exposed to one at all, suggesting that non-noticers have not represented anything about the location of the unexpected object. Consequently, we can use their location reports as a random-responding baseline.

Although nearly everything about the task and displays was identical across the two exposure duration conditions, they did differ in one potentially important way: the unexpected object's movement in the 5 s exposure condition spanned most of the display, near to both edges, whereas the motion in the 2.67 s condition was further from the edges. If proximity to the edges of the display influences noticing, that might interact with any effects of time on noticing or with location reports.

# 4. Experiment 2

In Experiment 2, we attempted to replicate and extend the findings of Experiment 1. We added an additional, shorter exposure time of 1.5 s to the 2.67 and 5 s conditions from Experiment 1. We also shifted the portion of the display the unexpected object traverses, so that it starts or ends at an edge of the display rather than being centred around fixation. The object could onset from either edge of the display and offset in the middle, or onset near the middle and offset at either edge. This allows us to examine whether there is any effect of an edge versus mid-display onset within an exposure duration. It also permits comparison of the pattern of noticing across durations when the objects onset from the same place on-screen versus when these objects onset at different points, revealing whether the pattern we observed in Experiment 1 changes with onset location.

This manipulation also provides a more robust test of the reliability of the location reports for all display durations. In Experiment 1, the left- and right-side onsets for the short display duration were both near fixation and not well-separated in space. In Experiment 2, even the shortest display duration conditions allow a comparison of an object that onsets at the far left or right edge of the display. Additionally, because there are conditions in which the onset position is in the middle of the display, far from the edges, it will be more apparent if subjects misreport the location of the object. For example, if they tend to extrapolate the location to the edge closest to where the object started (e.g. the left edge if the object travelled from left to right) irrespective of the actual onset location, then location reports should vary only with motion direction. If the misperception is milder, then we might expect a majority of the location reports to overshoot the onset point toward the edge closest to the start of the motion.

## 4.1. Methods

The materials and preregistration for this experiment are available at https://osf.io/jx9vs/. A demonstration of the task is available at simonslab.com/mot/temporal_mot_nc_demo.html.

### 4.1.1. Subjects

Using the procedures described in the General methods, we aimed to recruit 1200 subjects in order to end up with approximately 100 in each of six conditions after exclusions. We recruited 1273 in total.

### 4.1.2. Materials and procedure

The task, questionnaire and appearance of the unexpected object were identical to those of Experiment 1. Only the behaviour of the unexpected object differed.

In Experiment 2, there were six possible conditions representing a full crossing of exposure duration and onset behaviour. There were three different exposure times: 1.5 s, in which the unexpected object travelled 140 pixels; 2.67 s, in which it travelled 280 pixels and 5 s, in which it travelled 560 pixels. There were two different onset behaviours. In the 'edge onset' conditions, the unexpected object emerged from one edge of the display and offset behind an invisible occluder positioned 140, 280 or 560 pixels from the other edge. In the 'mid-display onset conditions, the objects onset from behind an invisible occluder positioned 140, 280 or 560 pixels into the display and offset at the far edge (figure 2); this mid-display-onset condition is similar to Experiment 1, in which objects also onset from behind invisible occluders positioned in the display and away from the edges. Whether the object travelled left-to-right or right-to-left was random for each subject.

## 4.2. Results and discussion

We excluded 517 subjects according to the criteria in the General methods (41% of our recruited subjects) and retained 756 in the analysis ($n = 107$ in the 1.5 s, edge-onset condition; $n = 127$ in the 1.5 s, mid-display-onset condition; $n = 129$ in the 2.67 s, edge-onset condition; $n = 136$ in the 2.67 s, mid-display-onset condition; $n = 132$ in the 5 s, edge-onset condition; and $n = 125$ in the 5 s, mid-display-onset condition).

### 4.2.1. Noticing

Subjects were coded as having noticed the object according to the same criteria used in Experiment 1.

Similar to the pattern observed in Experiment 1, reducing the exposure time had a small effect on noticing (figure 3): 41% of subjects (95% CI = 34.6–47.0) noticed the unexpected object in the 1.5 s exposure condition, 43.4% (95% CI = 37.0–49.8) noticed it in the 2.67 condition and 49.4% (95% CI = 43.6–55.6) noticed it in the 5 s condition.

There was also a small difference in noticing between objects that onset near the edge of the display and those that onset near the middle of the display. Edge-onset unexpected objects were noticed 47.6% of the time (95% CI = 42.7–52.4) and mid-display-onset objects were noticed 42.0% of the time (95% CI = 37.1–46.9). The size of this difference increased with longer exposure times, with a 0.2 (95% CI = −11.6–13.3) percentage point difference between edge and mid-display onsets in the 1.5 s condition, a 6.1 (95% CI = −6.0–18.0) percentage point difference in the 2.67 s condition, and a 9.0 (95% CI = −3.1–22.0) percentage point difference in the 5 s condition.

### 4.2.2. Location reports

As in Experiment 1, both noticers and non-noticers centred their vertical placements near the horizontal midline but with far more variability among the non-noticers (noticers $M$ = 294 pixels, s.d. = 78; non-noticers $M$ = 312 pixels, s.d. = 131). Noticers also placed the object on the onset side of fixation far more often than the offset side (85% of participants), whereas non-noticers were more evenly split (53% placed it on the onset side of fixation). Noticers generally placed the object close to the actual onset position, while non-noticer placements were more evenly distributed (table 3). This difference is particularly visible when examining the placements for objects that onset near the left or right edge of the display: Noticers placed the object near the edge of onset, but non-noticers did not (figure 4).

Figure 4 also shows that subjects did not dramatically misperceive the location of the unexpected object. Location reports clustered around onset, even when the onset point was positioned mid-display. Although there was variability in the reports, subjects did not extrapolate the location to the edge closest to where the motion started. Instead, their localizations were varied around the position where it actually first appeared.

The results of Experiment 2 replicate the pattern in Experiment 1. Dramatically reducing exposure time only modestly affected the probability of noticing. Even reducing exposure time by more than two-thirds only reduced noticing by 8 percentage points, suggesting that noticing occurs soon after onset if it is to occur at all; there is little additional benefit to having more exposure to the unexpected object.

Interestingly, there did appear to be a small effect of where the object onset on noticing rates. Subjects were slightly more likely to notice objects that onset at the edge than those that onset near the middle of the display. Given that the task requires monitoring bounces from the edge of the display, this difference in noticing might result from the deliberate allocation of attention to the edges. If so, that pattern provides further support for the idea that noticing happens soon after onset. Even though objects that onset near the middle of the display spend just as much time near the edges as those that onset at the edge, the heightened attention at the edge may be what enhances noticing for the edge-onset objects. No such attentional advantage for the mid-display onset objects may indicate that the window for noticing for objects that onset near the middle of the display might already be closed by the time those objects reach the edge of the display.

This small difference in noticing between edge- and mid-display-onset objects appeared to increase with exposure time. Mid-display-onset noticing rates barely increased with additional exposure time, rising from 41% in the 1.5 s condition to 45% in the 5 s condition. The edge-onset noticing rates increased by a greater amount, from 41% in the 1.5 s condition to 54% in the 5 s condition, although this increase is still small relative to the magnitude of the exposure time increase. However, there is a fair amount of variability in the estimate of the difference within each exposure condition; even the largest difference of 8 percentage points between the mid-display and edge onsets in the 5 s exposure condition has confidence intervals that include small negative differences. That said, the apparently larger effect in the 2.67 s and 5 s condition may indicate that the heightened attention to the edge of the display widens the window in which the unexpected object can be detected, and the 1.5 s condition is too brief to benefit; alternatively, it may be sufficiently close to the edge to receive the boost in both onset conditions.

This pattern might explain the higher noticing rates observed for slow (longer exposure) than fast (shorter exposure) objects in a previous study [10]. In that study, subjects monitored a horizontal line through the display and counted how often the attended objects touched the line. The unexpected object travelled horizontally through the display, parallel to this line, staying close to the focus of attention the entire time it was on-screen (including when it first appeared). The increased noticing

**Table 3.** Average Euclidean distance, in pixels, of subjects' location reports for the unexpected object to onset, fixation, and offset in Experiment 2. For each condition, onset type, and direction of motion, we averaged the distance between each individual location report and the location of onset in that condition, fixation, and the location for offset in that condition.

| noticed unexpected object | time on-screen | unexpected velocity | onset type | distance to fixation (s.d.) | distance to onset (s.d.) | distance to offset (s.d.) |
|---|---|---|---|---|---|---|
| no | 1.50 | right to left | edge | 176.1 (113.6) | 348.8 (152.0) | 237.6 (136.3) |
| no | 1.50 | right to left | mid-display | 178.3 (119.5) | 240.5 (117.1) | 342.4 (143.5) |
| no | 1.50 | left to right | edge | 167.8 (123.3) | 365.9 (165.9) | 252.8 (144.3) |
| no | 1.50 | left to right | mid-display | 208.6 (125.6) | 246.6 (138.2) | 341.9 (160.9) |
| no | 2.67 | right to left | edge | 171.1 (133.2) | 350.7 (158.9) | 182.4 (119.4) |
| no | 2.67 | right to left | mid-display | 159.3 (114.4) | 189.0 (111.3) | 406.7 (139.5) |
| no | 2.67 | left to right | edge | 147.2 (106.7) | 386.3 (141.7) | 172.0 (105.1) |
| no | 2.67 | left to right | mid-display | 184.5 (129.7) | 183.8 (110.7) | 329.3 (129.7) |
| no | 5.00 | right to left | edge | 202.1 (107.3) | 383.8 (167.3) | 277.5 (141.0) |
| no | 5.00 | right to left | mid-display | 118.9 (100.7) | 198.0 (75.5) | 416.2 (112.2) |
| no | 5.00 | left to right | edge | 163.0 (92.4) | 385.4 (133.8) | 250.3 (107.2) |
| no | 5.00 | left to right | mid-display | 190.0 (132.6) | 271.3 (112.2) | 400.7 (187.6) |
| yes | 1.50 | right to left | edge | 267.0 (112.8) | 202.8 (171.8) | 168.9 (133.7) |
| yes | 1.50 | right to left | mid-display | 174.2 (110.4) | 129.0 (67.7) | 210.6 (110.1) |
| yes | 1.50 | left to right | edge | 270.6 (82.0) | 103.7 (97.6) | 104.4 (59.9) |
| yes | 1.50 | left to right | mid-display | 184.8 (105.1) | 128.9 (101.4) | 213.3 (136.2) |
| yes | 2.67 | right to left | edge | 253.7 (101.0) | 130.3 (95.2) | 193.0 (89.1) |
| yes | 2.67 | right to left | mid-display | 129.2 (86.6) | 107.7 (84.3) | 284.5 (118.6) |
| yes | 2.67 | left to right | edge | 282.4 (93.4) | 126.0 (124.3) | 226.8 (76.7) |
| yes | 2.67 | left to right | mid-display | 147.5 (101.5) | 120.2 (106.4) | 282.4 (137.8) |
| yes | 5.00 | right to left | edge | 252.5 (111.8) | 218.2 (221.1) | 399.5 (152.0) |
| yes | 5.00 | right to left | mid-display | 163.7 (123.7) | 178.8 (104.5) | 459.2 (161.6) |
| yes | 5.00 | left to right | edge | 249.2 (93.1) | 119.7 (97.6) | 454.8 (98.7) |
| yes | 5.00 | left to right | mid-display | 200.0 (121.4) | 181.2 (112.8) | 486.3 (172.8) |

with longer exposure time in that experiment might result from the unexpected object onsetting near the locus of attention, just as we observed a greater impact of exposure time when the unexpected object first appeared at the edge of the display. Even in this case, however, the magnitude by which noticing increased was small despite the large increase in exposure time.

If exposure time has minimal impact on subjects' likelihood of noticing an unexpected object, does that imply that subjects in each condition also formed an equally accurate representation of the unexpected object regardless of how long they potentially could inspect it? Since detecting the unexpected object does not guarantee detailed or accurate encoding [29,30], might increased exposure time allow noticers to form a more accurate representation of the unexpected object's features? Experiment 3 addresses whether exposure time has an impact on how much detail subjects can encode about the object, even if the overall probability of noticing is relatively unaffected.

# 5. Experiment 3

Experiment 3 relied on a similar procedure to Experiment 1, using two exposure durations—1.5 and 5 s—with onset and offset locations near the middle of the display and centred around fixation. Instead of a grey cross, this time the unexpected object was a randomly chosen colour. Rather than a

forced-choice identification, subjects were asked to report the object's colour with a continuous colour slider. This allowed us to collect not just accuracy data, but also precision. Do subjects who have more exposure to the unexpected object also have a better representation of its features? That is, even though additional exposure time does not seem to affect noticing rates substantially, does it affect how much information about the unexpected object noticers are able to extract?

## 5.1. Methods

The materials and preregistration for this experiment are available at https://osf.io/wx5ua/. A demonstration of the task can be found at simonslab.com/mot/temporal_mot_col_demo.html.

### 5.1.1. Subjects

We aimed to recruit 1000 subjects in order to finish with 250 in each of two conditions, anticipating a 50% exclusion rate. We planned for a larger sample in each condition in order to better evaluate the precision of representations. We recruited 1004 subjects in total.

### 5.1.2. Materials and procedure

Experiment 3 used the same multiple object tracking task described in the General methods, but with a mid-grey (#808080) background instead of a light blue one. The fixation square was a 6 × 6 black square with a white, 2 pixel-wide border so that it would remain visible regardless of which colour of object passed behind it.

The unexpected object could be on-screen for either 1.5 or 5 s. Like Experiment 1 (and unlike Experiment 2), the unexpected object's motion was centred around fixation. It onset from behind an invisible occluder on one side of fixation, passed behind fixation and exited behind another invisible occluder on the opposite side of fixation, as in Experiment 1 (figure 2).

The unexpected object in this experiment again was a cross, and was one of 12 randomly chosen colours. The hues ranged from 0 to 330 in HSV (hue, saturation, value) space in 30° intervals, and the saturation and value were fixed at 50%. We ran two small-sample pilot studies with 41 and 62 subjects prior to the main experiment. The pilot procedure was identical to that of the main experiment but tested different colour values for the unexpected object in order to find a set of colours for which noticing was not at ceiling. In the first pilot, we used the same hues with saturation set to 100% and value set to 75%. Noticing was above 90% in both exposure conditions, so we reduced the intensity of the colours and ran the second pilot to verify that the change reduced overall noticing rates before proceeding to the primary experiment. The pilot data are available on the OSF page for this experiment.

Experiment 3 also changed how participants reported the colour of the unexpected object. Rather than selecting a colour from a drop-down menu of predetermined options, subjects matched the colour of the unexpected object with a slider. Clicking and dragging the marker on the slider adjusted the hue of a reference rectangle underneath it. The slider's saturation and value were set to 50%, so it was possible to exactly match the appearance of the unexpected object.

Subjects also completed a digital version of the Farnsworth D-15 task after the question about vision correction [26] (code adapted from a digital version of the task by Daniel Flück, https://www.color-blindness.com/color-arrangement-test/). Subjects had to order 15 coloured patches by chromaticity by dragging them into empty slots. They were provided with a fixed reference patch and told to complete the task by always selecting the next-most-similar colour and dropping it into the next available slot. This task was intended to identify colour deficiencies and monitor settings that interfered with the ability to discriminate colours. The procedure of Experiment 3 was otherwise the same as that described in the General methods.

## 5.2. Results and discussion

Prior to analysis, we excluded 516 subjects (51% of our sample) according to the criteria described in the General Methods. We retained data from 488 subjects ($n = 249$ in the 1.5 s condition and $n = 239$ in the 5 s condition).

### 5.2.1. Noticing

In Experiment 3, we did not classify subjects as noticers and non-noticers using the criteria from the first two experiments (i.e. reported noticing and correctly identified the shape and/or colour). Instead, given that the goal of this study was to examine precision in their representation of the unexpected object, we treated their self-report of noticing as evidence that they saw the unexpected object and then analysed their responses for the shape, colour and location of the unexpected object conditioned on whether they had reported seeing something new.

Replicating the main pattern from Experiments 1 and 2, self-reported noticing rates were similar across the two exposure conditions (figure 3). 61.5% (95% CI = 55.4–67.5) of subjects reported noticing something new in the 1.5 s condition, and 64.9% (95% CI = 58.6–70.7) of subjects reported noticing something new in the 5 s condition for a difference of −3.4 percentage points (95% CI = −12.0–5.1).

### 5.2.2. Feature reporting accuracy

Self-reported noticers in both exposure conditions were highly accurate at reporting the shape of the unexpected object. Of the self-reported noticers in the 1.5 s exposure condition, 87.6% (95% CI = 82.4–92.8) correctly reported the shape of the unexpected object, and 88.4% (95% CI = 83.2–92.9) of them did so in the 5 s condition. Non-noticers in the 1.5 s and 5 s condition were approximately at chance levels (11.1%) in selecting the unexpected object's shape correctly (1.5 s: 10.4%, 95% CI = 4.2–16.7; 5 s: 8.3%, 95% CI = 2.4–14.3).

Self-reported noticers in both conditions were extremely accurate at reporting the colour, with a circular mean error of −1.1° (angular deviation = 27.2) in the 1.5 s exposure condition and a circular mean error of −1.9° (angular deviation = 29.2) in the 5 s condition (figure 4). The difference in means between the two conditions was 0.83° (95% CI = −4.3–5.4), and the ratio of the circular variance between the two conditions was 0.87 (95% CI = 0.45–1.66). Thus, the two conditions appear to be not just equally accurate at reporting the colour, but equally precise. Non-noticers, in contrast, had a mean circular error of −109.9° (angular deviation = 79.9) in the 1.5 s condition and 171.8 (angular deviation = 79.3) in the 5 s condition (figure 5). If we generate 1000 samples of 100 subjects selecting a colour completely randomly from any point on the colour wheel, the mean angular deviation of the error would be 77.4° (95% CI = 72.6–80.4). In other words, the variability in colour selection shown by non-noticers is what we would expect from chance responding.

### 5.2.3. Location reports

Consistent with the results of Experiments 1 and 2, self-reported noticers localized the unexpected object near its onset location on average (figure 4). Noticers had a tighter spread to their vertical placements than did non-noticers, as in Experiments 1 and 2, placing their objects at an average of 286 pixels (s.d. = 75) compared to 295 (s.d. = 159) for non-noticers. Noticers placed the object on the onset side of fixation 70% of the time; non-noticers did so 45% of the time. Noticers generally placed the object closest to the onset location, whereas non-noticer localizations were more evenly distributed (table 4). The tendency to localize the unexpected object near the onset location is easier to see in the 5 s condition where there is more space between onset and fixation than in the 1.5 s condition, when fixation, onset and offset were more compressed.

Experiment 3 closely replicates the pattern of results observed in the previous experiments. First, dramatically reduced exposure time to the unexpected object had comparatively little effect on the probability that subjects would notice it. Second, subjects tended to report noticing the unexpected object near its onset location. An appreciable proportion of noticing events apparently occur soon after the unexpected object appears.

The results of Experiment 3 further suggest that not only is there minimal impact of exposure time on the likelihood of noticing the unexpected object, but also minimal impact on the precision of the representation of that object. Subjects who noticed the unexpected object were equally accurate at reporting the object's shape and colour whether they were exposed to it for 1.5 s or 5 s. The extra exposure time did not improve accuracy or precision. Indeed, there was almost no room to improve: the circular mean of subjects' reports of the colour in the 1.5 s condition were already nearly perfect. However, the accuracy for reporting the shape, which was not at ceiling and could have reflected a difference, was also the same across the two exposure conditions. It appears that subjects in both conditions had equally accurate representations of the unexpected object despite a large difference in

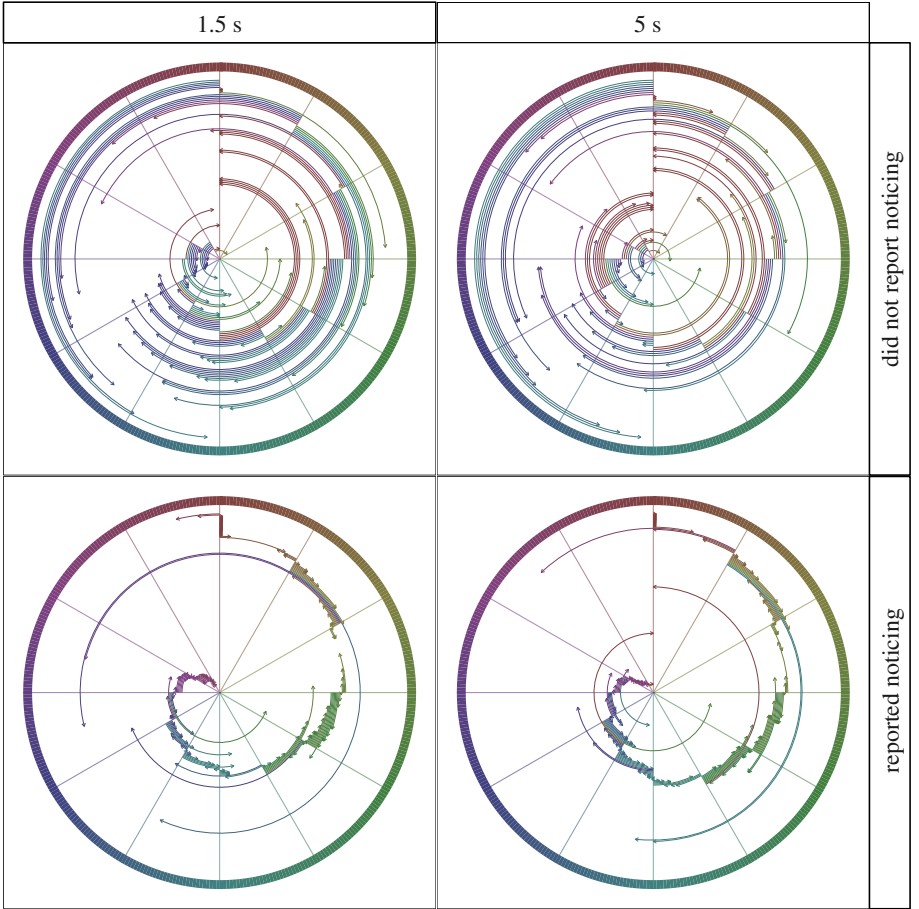

**Figure 5.** Raw colour error for each subject. The line segment begins at the true hue of the unexpected object, indicated by the spokes in the wheel. The line ends with the arrow pointing to the subjects' reported hue, and the line segment is coloured to match the reported hue.

**Table 4.** Average Euclidean distance, in pixels, of subjects' location reports for the unexpected object to onset, fixation and offset in Experiment 3. For each condition and direction of motion, we averaged the distance between each individual location report and the location of onset in that condition, fixation and the location for offset in that condition.

| noticed unexpected object | time on-screen | unexpected velocity | distance to fixation | distance to onset | distance to offset |
|---|---|---|---|---|---|
| no | 1.5 | right to left | 234.7 (148.7) | 246.4 (151.4) | 247.0 (139.9) |
| no | 1.5 | left to right | 199.0 (148.8) | 214.0 (137.1) | 215.9 (148.3) |
| no | 5.0 | right to left | 174.6 (132.2) | 307.2 (160.3) | 338.9 (134.0) |
| no | 5.0 | left to right | 223.7 (139.8) | 352.0 (135.7) | 339.4 (197.2) |
| yes | 1.5 | right to left | 84.1 (89.6) | 106.1 (90.7) | 115.3 (85.0) |
| yes | 1.5 | left to right | 81.7 (103.8) | 104.6 (93.8) | 117.8 (105.3) |
| yes | 5.0 | right to left | 218.9 (108.4) | 157.1 (136.7) | 459.3 (148.2) |
| yes | 5.0 | left to right | 244.1 (112.6) | 144.7 (126.2) | 492.7 (148.0) |

exposure time. By contrast, non-noticers were inaccurate in both their colour and shape reports. Although we did not observe a benefit of exposure time on feature reporting accuracy, our unexpected object had only two features (colour and shape), and it is possible that such a benefit would emerge with a more complex object.

# 6. General discussion

Across three experiments, substantially reducing the amount of time an unexpected object was on-screen, by 50% or even 70%, had only a modest impact on the proportion of subjects noticing it. The largest difference in noticing as a function of exposure time was for unexpected objects that onset from the edge of the display in Experiment 2, in which there was a 12.7 percentage point difference between the 5 s and 1.5 s exposure durations.

The window for noticing an unexpected object appears to be brief relative to the amount of time it is visible on-screen. Even though subjects have more opportunity to detect the unexpected object the longer it remains on-screen, the vast majority of noticing events occur in the first 1.5 s or not at all. This pattern, replicated in all three experiments, indicates that unexpected objects are not noticed as a result of a gradual accumulation of signal across the entire exposure duration, but more as a result of a rapid process concentrated early in the unexpected object's lifespan.

The results from subjects' reports of when they first noticed the unexpected object help narrow down when these noticing events occur. Subjects who noticed the unexpected object tended to report first seeing it near the onset location in all conditions. The onset event itself might trigger noticing, with a brief window of opportunity for detection shutting rapidly afterward. The abrupt appearance of a new object in the absence of other events or distractions can provide a strong attention signal in other tasks [31]. However, the effectiveness of such onsets in capturing attention is reduced when they coincide with other dynamic events [32].

Experiment 3 further revealed that not only does exposure time have little impact on noticing, it also seems to have little impact on the quality of the representation of the unexpected object's features. This presents the counterintuitive possibility that if an object is noticed, it is also represented at the highest precision it could be. More exposure to the object after the initial noticing event did not further improve the representation. If so, detection and representation of the unexpected object might be an all-or-nothing process. It either happens in its entirety within a short window after onset, or not at all. Alternatively, detection and representation may be distinct from each other, but on a time scale shorter than 1.5 s.

What precisely unfolds in those initial 1.5 s remains an avenue for exploration. We can be confident that the early noticing we observed is not due to a ceiling effect. Noticing rates hovered around 50%, meaning that the unexpected object was not so conspicuous that everyone noticed it. Many people missed the object, but those who noticed it appeared to do so rapidly. The onset event itself, and the associated visual change, might trigger noticing of the unexpected object. Or, once an additional object appears, the attentional system might rapidly accumulate signal which passes some threshold for detection with a given probability (meaning it is noticed) or does not (meaning it is missed). There may even be competition between the signal generated by the new object and suppression of task-irrelevant information, and this suppression of irrelevant information could affect the likelihood of detecting the object after onset.

The time intervals used in the three experiments and the reports of the unexpected object's location are not precise enough to definitively support one of these models. However, these results can rule out other possible accounts of how noticing unfolds in sustained inattentional blindness. Noticing was not triggered by offset events or by the unexpected object crossing fixation, and noticing is neither a slow process that unfolds over time, nor a slow accumulation of evidence that accelerates the longer the unexpected object is visible. Rather, noticing in these sustained inattentional blindness tasks may largely be a process that happens almost immediately or not at all.

# 7. Constraints on generality

We expect these results to generalize to similar tasks and to be robust to arbitrary choices about the appearance and behaviour of the stimuli and display, provided that overall noticing levels are not driven to ceiling or floor. Other studies have used sustained inattentional blindness tasks like these across a range of settings (laboratory, public, online), so we expect our results to generalize to adult populations with normal or corrected-to-normal visual acuity in both online and in-person settings.

Our results in Experiment 2 suggest that differences in noticing rates as a function of spatial attention probably would not interact with the exposure duration effects we examined. Similarity effects, wherein objects similar to the attended set tend to be noticed at high rates, whereas objects in the ignored set tend to be noticed at much lower rates [6], might interact with the effects of exposure duration we observed in these experiments.

Precision for colour selection was near ceiling in Experiment 3, but that does not necessarily mean that exposure time has no effect on how accurately subjects can report features. Perhaps the task used in Experiment 3 was too easy for subjects, and a more difficult task would reveal effects of exposure; perhaps a more complex object with more features than shape and colour would benefit from additional time, even though the simple object we used did not. Additionally, it is possible that much longer exposures would yield more advantage and that 5 s is not enough additional time to reveal a benefit of additional exposure. Exposure time has an impact on encoding and representation in other tasks, such as change blindness [33]. These effects may still be present in inattentional blindness as well, but may either saturate at 1.5 s, require a harder task, require even *more* time to reveal themselves, or some combination of these factors.

Our results might also be limited to simple tasks and displays and might not generalize to more real-world or video-based tasks, where the content and interaction between the action of the video and the unexpected object can affect the time course of noticing [8].

Ethics. The need for signed consent was waived by the Institutional Review Board of the University of Illinois at Urbana-Champaign due to the minimal risk nature of the experiment (protocol no. 09441). Subjects were informed prior to participation of the nature of the study, that their data would remain anonymous and no attempt would be made to identify them, and that participation was voluntary and could be terminated at any point. They were also provided with contact information for author K.W. and the UIUC IRB board in the event that they had concerns or complaints.

Data accessibility. All data, experimental materials and analysis code is available on this study's OSF project page at https://osf.io/gb6v5/ [15].

Authors' contributions. K.W. and D.J.S. jointly planned and designed the experiments. K.W. coded the experiments and analysis scripts, oversaw data collection, conducted the analysis and drafted the manuscript. Both authors critically edited and revised the manuscript and approved the final version for publication.

Competing interests. We declare we have no competing interests.

Funding. K.W. is supported by a National Science Foundation Graduate Research Fellowship.

Acknowledgements. The authors thank Daniel Flück for his help with the Farnsworth D-15 code and the reviewers of this manuscript for their helpful comments.

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
