## [Reviewer comments · Royal Society Open Science]

Review History

RSOS-191333.R0 (Original submission)

Review form: Reviewer 1 (Carina Kreitz)

Is the manuscript scientifically sound in its present form?

Yes

Are the interpretations and conclusions justified by the results?

Yes

Is the language acceptable?

Yes

Do you have any ethical concerns with this paper?

No

Have you any concerns about statistical analyses in this paper?

No

Recommendation?

Accept with minor revision (please list in comments)

Comments to the Author(s)

The authors' investigate noticing in an sustained inattentional blindness paradigm depending on the time the unexpected object is visible in the display. They do so in three carefully designed online experiments that in combination allow new insights into the underlying mechanism of this failure of awareness.

I think that the manuscript constitutes a valuable addition to the literature and should be published provided some revision.

Comments and suggestions:

- (1) My main comment is that the authors are not the first to systematically investigate the exposure time on noticing. I did that several years ago (Kreitz, C., Furley, P., & Memmert, D. (2016). Inattentional blindness is influenced by exposure time not motion speed. *Quarterly Journal of Experimental Psychology*, 69(3), 495-505). Also, Beanland and Pammer used different exposure times in one of their experiments (Beanland, V., & Pammer, K. (2010). Looking without seeing or seeing without looking? Eye movements in sustained inattentional blindness. *Vision Research*, 50(10), 977-988.). Kreitz et al. (2016) found that exposure time in a similar paradigm (adapted from Most; judging whether the unexpected object crossed a horizontal line in the vertical middle of the display) significantly and strongly influenced detection rates (20-25% more noticing). So, the author's should integrate these findings into their manuscript; what sets their aims and design apart? And, how can the findings be integrated (as some inferences are completely opposed)? I believe that the manuscript nevertheless valuably adds to the literature (WHEN people notice is completely new for example; and they also use a slightly different task design), but some additional effort is needed to integrate those former experiments.
- (2) In general, the manuscript only scarcely cites adjacent literature. At some points of the manuscript this is needed, though, in my opinion. For example, the authors discuss the possibility that transients (in combination with the spatial focus of attention) determined detection rates; what does the literature on transients (e.g., from change blindness or attentional capture) say? Also, for example, p.21, l.14: reference for "since detecting the unexpected object does not guarantee detailed or accurate encoding"; or: last paragraph p.27, references from other failures of awareness?
- (3) Why do the authors not use a full-attention trial as control? At least they should explain their choice as it is common in inattentional blindness literature to make sure that each observer is capable of perceiving the unexpected object. And I believe that this is especially important in an online study; participants have different devices with different monitors, different brightness levels etc.
- (4) The thoughts of the authors on the misperception of location due to motion are interesting. I wonder whether the pattern of results could stem from that? If we think drastically that location is perceived to the side the object comes from, the pattern of localization reports seems plausible independent of actual location of awareness. I guess what speaks against that are the findings from the 1.5s condition in Exp 3, as localization reports should be far more lateralized there, too, if the alternative explanation were true. Perhaps the authors can go a little bit deeper into this argumentation.
- (5) In general, some information is given multiple times. Perhaps some sentences could be shortened (e.g., p.23, p.28, second paragraph)
- (6) The authors might discuss that their findings on the location (and, thus, time) of noticing is not a ceiling effect (which could be an alternative explanation for their findings; it is so easy to notice the stimulus that all observers do so in the first second) as there are a lot of non-noticers, too.

Minor comments:

- (7) I am no native speaker, but using “onset” and “offset” as verb seems odd.
- (8) I really like the figures the authors use to illustrate their hypotheses, design, and results. I do understand why the authors integrated all three experiments in one figure, however, I found it really impractical and had to go back a lot of times. Perhaps this problem vanishes in the final typesetting. Another solution could be that each experiment gets one figure with method and both results?
- (9) The authors use some abbreviations that are never explained. For example, “IRB”, “EKP”, “HSV” (I know what EKP is, but not all readers might..?). “HIT” is explained when it is used the second time, not first time.
- (10) p. 7, l.6; Kreitz et al. suggested such a stochastic process in their paper.
- (11) Typo: p.10, l.22: should be “than” instead of “that”
- (12) Typo: p.12, l.6: “that the these”
- (13) I did not get the vertical colored lines in Figure 4. Are they correct?
- (14) I do not get the term “floating” in contrast to “edge onset”. Perhaps there is a more intuitive term?
- (15) P. 20/21: I do not fully get the paragraph that goes from p.20 to 21; rephrase. Perhaps some amount of this variability is due to random attentional shifts?
- (16) P. 21, l.9: make clearer that you mean the onset when speaking about the “narrow window of time”
- (17) P.21: what do you mean by “trajectories that began and ended near the middle of the display”? Is that correct?

Review form: Reviewer 2

Is the manuscript scientifically sound in its present form?

Yes

Are the interpretations and conclusions justified by the results?

Yes

Is the language acceptable?

Yes

Do you have any ethical concerns with this paper?

No

Have you any concerns about statistical analyses in this paper?

No

Recommendation?

Accept with minor revision (please list in comments)

Comments to the Author(s)

Summary:

When people notice an unexpected event or object in an inattentional blindness paradigm, do they notice immediately, or do they need sufficient time to notice and for the representation of the event or object to form? In three experiments, the authors convincingly show that "noticers" notice unexpected objects right away, and extra exposure time only contributing a small effect. When the object first appears close to the edge of the screen (where participants are attending for

their primary task), people are more likely to notice it compared to when it appears closer to fixation. Finally, when people notice the objects, they immediately or very quickly represent it fully, as demonstrated by highly precise color recall.

Comments:

This paper is exceedingly clear: the question is clear (and interesting!), the hypotheses are clear, the writing is great, and the data are convincing, and the interpretation is warranted. I have no major comments, only a few optional suggestions or comments.

Minor comments:

Page 11, Table 1: Can you include the total number of people actually used in the experiments, to compare to the number excluded?

Also, about exclusions: These exclusions are very conservative, which ensures high quality of the results, but I'm wondering about the vision correction question. Do you have any sense of what correction they aren't using (and whether they'd actually need it for sitting at a computer)? I'm just surprised, it seems like a lot of people. Also, same for the technical issues: what problems did people have with it?

Page 17, line 11: The "floating" onset is what was used in Experiment 1, correct? Might good to specify that (e.g. "(like in Experiment 1)"), so readers know it's not a new manipulation.

Page 20, Line 28: If the participants had been monitoring mid-line crossings, which is another standard inattentive blindness primary task, this would allow you to determine if the edge-advantage was due to attention, or something particular about edges per se.

Review form: Reviewer 3

Is the manuscript scientifically sound in its present form?

Yes

Are the interpretations and conclusions justified by the results?

Yes

Is the language acceptable?

Yes

Do you have any ethical concerns with this paper?

No

Have you any concerns about statistical analyses in this paper?

No

Recommendation?

Accept with minor revision (please list in comments)

Comments to the Author(s)

I read this this paper with great interest. Indeed, if the research question can seem "simple", it is very interesting and can provide a real contribution to the field.

In a general fashion, I have a very positive opinion about this paper. I really appreciated that it complies with new research practices and open science: All studies are pre-registered, high-powered, and materials, data and R scripts are available on osf. Note that the link to the pre-registration document is broken, so I was unable to check whether or not the performed analyses are the same as the pre-registered ones, etc (but I'm pretty confident they are).

Below are questions or minor concerns I had while reviewing the paper.

Why was the offset fixed at 2 s. before the end of the trial?

More generally, I think that the authors could justify more the time frames they used.

p. 11, why 2,7 s. for short exposure and 5 s. for long exposure? For instance, I think that the authors could justify the 5s exposure time in a better way than "a 5s exposure is typical for this sort of sustained inattentive blindness task". I also wondered why the authors chose to use 2 exposure times and not 3? (even if I found some answer to this question in exp. 2).

p 16 " in general, though, subjects who noticed the unexpected object positioned it close to its actual onset location with some precision ". Wouldn't it be possible to estimate more precisely the accuracy of the detection by calculating an error score for example?

In exp 3, I think that the conclusions of the study about the quality of the representation should more take in account that the unexpected object is very simple and has few features (a basic shape and one color). Conclusions about exposure times may be different with a more complex object.

p.27. The authors propose that "the onset event itself might trigger noticing". What if something in the unexpected object could grab attention (blinking, changing color)?

Finally, was there a deterioration in the primary task (i.e., counting the number of bounces) when participants detected the unexpected event?

Decision letter (RSOS-191333.R0)

28-Sep-2019

Dear Ms Wood

On behalf of the Editors, I am pleased to inform you that your Manuscript RSOS-191333 entitled "Now or never: Noticing occurs early in sustained inattentive blindness" has been accepted for publication in Royal Society Open Science subject to minor revision in accordance with the referee suggestions. Please find the referees' comments at the end of this email.

The reviewers and handling editors have recommended publication, but also suggest some minor revisions to your manuscript. Therefore, I invite you to respond to the comments and revise your manuscript.

- Ethics statement

- Data accessibility

If you wish to submit your supporting data or code to Dryad (<http://datadryad.org/>), or modify your current submission to dryad, please use the following link:
<http://datadryad.org/submit?journalID=RSOS&manu=RSOS-191333>

- Competing interests

- Authors' contributions

- Acknowledgements

- Funding statement

Because the schedule for publication is very tight, it is a condition of publication that you submit the revised version of your manuscript before 07-Oct-2019. Please note that the revision deadline

will expire at 00.00am on this date. If you do not think you will be able to meet this date please let me know immediately.

If your manuscript is newly submitted and subsequently accepted for publication, you will be asked to pay the article processing charge, unless you request a waiver and this is approved by

Royal Society Publishing. You can find out more about the charges at <http://rsos.royalsocietypublishing.org/page/charges>. Should you have any queries, please contact openscience@royalsociety.org.

on behalf of Dr Narayanan Srinivasan (Associate Editor) and Essi Viding (Subject Editor)
openscience@royalsociety.org

Associate Editor Comments to Author (Dr Narayanan Srinivasan):

Associate Editor: 1

Comments to the Author:

Three reviewers have now commented on the paper. They are generally positive. Authors are requested to address all the comments and submit their revision.

Associate Editor: 2

Comments to the Author:

(There are no comments.)

Reviewer comments to Author:

Reviewer: 1

Comments to the Author(s)

The authors' investigate noticing in an sustained inattentional blindness paradigm depending on the time the unexpected object is visible in the display. They do so in three carefully designed online experiments that in combination allow new insights into the underlying mechanism of this failure of awareness.

I think that the manuscript constitutes a valuable addition to the literature and should be published provided some revision.

Comments and suggestions:

(1) My main comment is that the authors are not the first to systematically investigate the exposure time on noticing. I did that several years ago (Kreitz, C., Furley, P., & Memmert, D. (2016). Inattentional blindness is influenced by exposure time not motion speed. *Quarterly Journal of Experimental Psychology*, 69(3), 495-505). Also, Beanland and Pammer used different exposure times in one of their experiments (Beanland, V., & Pammer, K. (2010). Looking without seeing or seeing without looking? Eye movements in sustained inattentional blindness. *Vision Research*, 50(10), 977-988.). Kreitz et al. (2016) found that exposure time in a similar paradigm (adapted from Most; judging whether the unexpected object crossed a horizontal line in the vertical middle of the display) significantly and strongly influenced detection rates (20-25% more noticing). So, the author's should integrate these findings into their manuscript; what sets their aims and design apart? And, how can the findings be integrated (as some inferences are

completely opposed)? I believe that the manuscript nevertheless valuably adds to the literature (WHEN people notice is completely new for example; and they also use a slightly different task design), but some additional effort is needed to integrate those former experiments.

(2) In general, the manuscript only scarcely cites adjacent literature. At some points of the manuscript this is needed, though, in my opinion. For example, the authors discuss the possibility that transients (in combination with the spatial focus of attention) determined detection rates; what does the literature on transients (e.g., from change blindness or attentional capture) say? Also, for example, p.21, l.14: reference for “since detecting the unexpected object does not guarantee detailed or accurate encoding”; or: last paragraph p.27, references from other failures of awareness?

(3) Why do the authors not use a full-attention trial as control? At least they should explain their choice as it is common in inattention blindness literature to make sure that each observer is capable of perceiving the unexpected object. And I believe that this is especially important in an online study; participants have different devices with different monitors, different brightness levels etc.

(4) The thoughts of the authors on the misperception of location due to motion are interesting. I wonder whether the pattern of results could stem from that? If we think drastically that location is perceived to the side the object comes from, the pattern of localization reports seems plausible independent of actual location of awareness. I guess what speaks against that are the findings from the 1.5s condition in Exp 3, as localization reports should be far more lateralized there, too, if the alternative explanation were true. Perhaps the authors can go a little bit deeper into this argumentation.

(5) In general, some information is given multiple times. Perhaps some sentences could be shortened (e.g., p.23, p.28, second paragraph)

(6) The authors might discuss that their findings on the location (and, thus, time) of noticing is not a ceiling effect (which could be an alternative explanation for their findings; it is so easy to notice the stimulus that all observers do so in the first second) as there are a lot of non-noticers, too.

Minor comments:

(7) I am no native speaker, but using “onset” and “offset” as verb seems odd.

(8) I really like the figures the authors use to illustrate their hypotheses, design, and results. I do understand why the authors integrated all three experiments in one figure, however, I found it really impractical and had to go back a lot of times. Perhaps this problem vanishes in the final typesetting. Another solution could be that each experiment gets one figure with method and both results?

(9) The authors use some abbreviations that are never explained. For example, “IRB”, “EKP”, “HSV” (I know what EKP is, but not all readers might..?). “HIT” is explained when it is used the second time, not first time.

(10) p. 7, l.6; Kreitz et al. suggested such a stochastic process in their paper.

(11) Typo: p.10, l.22: should be “than” instead of “that”

(12) Typo: p.12, l.6: “that the these”

(13) I did not get the vertical colored lines in Figure 4. Are they correct?

(14) I do not get the term “floating” in contrast to “edge onset”. Perhaps there is a more intuitive term?

(15) P. 20/21: I do not fully get the paragraph that goes from p.20 to 21; rephrase. Perhaps some amount of this variability is due to random attentional shifts?

(16) P. 21, l 9: make clearer that you mean the onset when speaking about the “narrow window of time”

(17) P.21: what do you mean by “trajectories that began and ended near the middle of the display”? Is that correct?

Reviewer: 2

Comments to the Author(s)

Summary:

When people notice an unexpected event or object in an inattentional blindness paradigm, do they notice immediately, or do they need sufficient time to notice and for the representation of the event or object to form? In three experiments, the authors convincingly show that "noticers" notice unexpected objects right away, and extra exposure time only contributing a small effect. When the object first appears close to the edge of the screen (where participants are attending for their primary task), people are more likely to notice it compared to when it appears closer to fixation. Finally, when people notice the objects, they immediately or very quickly represent it fully, as demonstrated by highly precise color recall.

Comments:

This paper is exceedingly clear: the question is clear (and interesting!), the hypotheses are clear, the writing is great, and the data are convincing, and the interpretation is warranted. I have no major comments, only a few optional suggestions or comments.

Minor comments:

Page 11, Table 1: Can you include the total number of people actually used in the experiments, to compare to the number excluded?

Also, about exclusions: These exclusions are very conservative, which ensures high quality of the results, but I'm wondering about the vision correction question. Do you have any sense of what correction they aren't using (and whether they'd actually need it for sitting at a computer)? I'm just surprised, it seems like a lot of people. Also, same for the technical issues: what problems did people have with it?

Page 17, line 11: The "floating" onset is what was used in Experiment 1, correct? Might good to specify that (e.g. "(like in Experiment 1)"), so readers know it's not a new manipulation.

Page 20, Line 28: If the participants had been monitoring mid-line crossings, which is another standard inattentional blindness primary task, this would allow you to determine if the edge-advantage was due to attention, or something particular about edges per se.

Reviewer: 3

Comments to the Author(s)

I read this this paper with great interest. Indeed, if the research question can seem "simple", it is very interesting and can provide a real contribution to the field.

In a general fashion, I have a very positive opinion about this paper. I really appreciated that it complies with new research practices and open science: All studies are pre-registered, high-powered, and materials, data and R scripts are available on osf. Note that the link to the pre-registration document is broken, so I was unable to check whether or not the performed analyses are the same as the pre-registered ones, etc (but I'm pretty confident they are).

Below are questions or minor concerns I had while reviewing the paper.

Why was the offset fixed at 2 s. before the end of the trial?

More generally, I think that the authors could justify more the time frames they used.

p. 11, why 2,7 s. for short exposure and 5 s. for long exposure? For instance, I think that the authors could justify the 5s exposure time in a better way than "a 5s exposure is typical for this sort of sustained inattention blindness task". I also wondered why the authors chose to use 2 exposure times and not 3? (even if I found some answer to this question in exp. 2).

p 16 " in general, though, subjects who noticed the unexpected object positioned it close to its actual onset location with some precision ". Wouldn't it be possible to estimate more precisely the accuracy of the detection by calculating an error score for example?

In exp 3, I think that the conclusions of the study about the quality of the representation should more take in account that the unexpected object is very simple and has few features (a basic shape and one color). Conclusions about exposure times may be different with a more complex object.

p.27. The authors propose that "the onset event itself might trigger noticing". What if something in the unexpected object could grab attention (blinking, changing color)?

Finally, was there a deterioration in the primary task (i.e., counting the number of bounces) when participants detected the unexpected event?

Author's Response to Decision Letter for (RSOS-191333.R0)

See Appendix A.

Decision letter (RSOS-191333.R1)

24-Oct-2019

Dear Ms Wood,

I am pleased to inform you that your manuscript entitled "Now or never: Noticing occurs early in sustained inattention blindness" is now accepted for publication in Royal Society Open Science.

Royal Society Open Science operates under a continuous publication model (<http://bit.ly/cpFAQ>). Your article will be published straight into the next open issue and this will be the final version of the paper. As such, it can be cited immediately by other researchers.

As the issue version of your paper will be the only version to be published I would advise you to check your proofs thoroughly as changes cannot be made once the paper is published.

Kind regards,

on behalf of Dr Narayanan Srinivasan (Associate Editor) and Professor Essi Viding (Subject Editor)
openscience@royalsociety.org

Appendix A

Dear Editors,

We thank the editors and the reviewers for their consideration and helpful comments on the manuscript. We revised the manuscript based on the feedback we received, and we hope that you will find this revised version of the manuscript to be suitable for publication.

Below, we have reproduced the reviewer comments (indented and italicized) along with our responses (out-dented). Reviewer 1 suggested breaking the methods and results figures up due to the challenges of flipping back and forth across experiments, but we hope that the formatted manuscript will minimize that issue. We can reorganize these figures if the editor feels it will make it easier for readers, although we arranged the figures to allow easy comparison across experiments and would like to prioritize that if it won't cause issues. Please let us know if you feel it would be better to rearrange them.

Sincerely,

Katherine Wood

Reviewer comments to Author:

Reviewer: 1

Comments to the Author(s)

The authors' investigate noticing in an sustained inattentional blindness paradigm depending on the time the unexpected object is visible in the display. They do so in three carefully designed online experiments that in combination allow new insights into the underlying mechanism of this failure of awareness.

I think that the manuscript constitutes a valuable addition to the literature and should be published provided some revision.

Comments and suggestions:

(1) My main comment is that the authors are not the first to systematically investigate the exposure time on noticing. I did that several years ago (Kreitz, C., Furley, P., & Memmert, D. (2016). Inattentional blindness is influenced by exposure time not motion speed. Quarterly Journal of Experimental Psychology, 69(3), 495-505). Also, Beanland and Pammer used different exposure times in one of their experiments (Beanland, V., & Pammer, K. (2010). Looking without seeing or seeing without looking? Eye movements in sustained inattentional blindness. Vision Research, 50(10), 977-988.). Kreitz et al. (2016) found that exposure time in a similar paradigm (adapted from Most; judging whether the unexpected object crossed a horizontal line in the vertical middle of the display) significantly and strongly influenced detection rates (20-25% more noticing). So, the author's should integrate these findings into their manuscript; what sets their aims and design apart? And, how can the findings be integrated (as some inferences are completely opposed)? I believe that the manuscript nevertheless valuably adds to the literature (WHEN people notice is completely new for example; and they also use a slightly different task design), but some additional effort is needed to integrate those former experiments.

We apologize for the omission, and have added additional discussion of these experiments to our Introduction, General Discussion, and particular experiment discussion sections where relevant. We review the findings of these papers, discuss critical design differences, and discuss the apparently opposed conclusions of these experiments. Our results in Experiment 2 are consistent with the results of Kreitz et al., 2016—we too observed a small increase in noticing rates with additional exposure time, but only when the unexpected object's onset point was in an attentionally relevant area at the edge of the display. In the Kreitz et al. experiment, the unexpected object always traveled quite close to the horizontal line that subjects were monitoring for their primary task. The effect of exposure time observed in those experiments is consistent with our results because that object also onset in an area of attentional relevance.

(2) In general, the manuscript only scarcely cites adjacent literature. At some points of the manuscript this is needed, though, in my opinion. For example, the authors discuss the possibility that transients (in combination with the spatial focus of attention) determined detection rates; what does the literature on transients (e.g., from change blindness or attentional capture) say? Also, for example, p.21, l.14: reference for "since detecting the unexpected object does not guarantee detailed or accurate encoding"; or: last paragraph p.27, references from other failures of awareness?

We have added additional supporting citations throughout the manuscript.

(3) Why do the authors not use a full-attention trial as control? At least they should explain their choice as it is common in inattention blindness literature to make sure that each observer is capable of perceiving the unexpected object. And I believe that this is especially important in an online study; participants have different devices with different monitors, different brightness levels etc.

Our previous experience in using full-attention trials in these experiments is that they increase exclusions without changing the pattern of results, or sometimes even without changing the point estimates. People who perform well on the primary task, report noticing the object, and correctly respond with its features can fail the full attention trial, which suggests that the full attention trial is not a reliable indicator of overall perceptibility in this context (at least, it doesn't add much to the other indicators that participants are engaged in the task). Unlike the original Mack & Rock studies that presented brief exposures to the unexpected object, there's little concern that these well-above-threshold stimuli are visible. Indeed, other researchers using online experiments to study inattention blindness do not routinely employ full-attention trials (e.g. Ward & Scholl, 2015; Drew & Stothart, 2016). We have added a note to this effect to the manuscript.

(4) The thoughts of the authors on the misperception of location due to motion are interesting. I wonder whether the pattern of results could stem from that? If we think drastically that location is perceived to the side the object comes from, the pattern of localization reports seems plausible independent of actual location of awareness. I guess what speaks against that are the findings from the 1.5s condition in Exp 3, as localization reports should be far more lateralized there, too, if the alternative explanation were true. Perhaps the authors can go a little bit deeper into this argumentation.

This is a worthwhile suggestion. We have added additional discussion of this point to the introduction and discussion of Experiment 2, in which the onset points move the most dramatically and the localization performance follow them in a way that argues against extrapolation toward the starting side of the motion.

(5) *In general, some information is given multiple times. Perhaps some sentences could be shortened (e.g., p.23, p.28, second paragraph)*

We have edited the manuscript throughout for clarity and concision, particularly in the areas indicated.

(6) *The authors might discuss that their findings on the location (and, thus, time) of noticing is not a ceiling effect (which could be an alternative explanation for their findings; it is so easy to notice the stimulus that all observers do so in the first second) as there are a lot of non-noticers, too.*

This is a good suggestion, and we have added discussion of this point to the general discussion.

Minor comments:

(7) *I am no native speaker, but using “onset” and “offset” as verb seems odd.*

While this usage is not necessarily routine, it is used in the inattentive blindness literature (e.g. Most et al., 2005).

(8) *I really like the figures the authors use to illustrate their hypotheses, design, and results. I do understand why the authors integrated all three experiments in one figure, however, I found it really impractical and had to go back a lot of times. Perhaps this problem vanishes in the final typesetting. Another solution could be that each experiment gets one figure with method and both results?*

See our note to the editor above.

(9) *The authors use some abbreviations that are never explained. For example, “IRB”, “EKP”, “HSV” (I know what EKP is, but not all readers might..?). “HIT” is explained when it is used the second time, not first time.*

We have clarified the abbreviations throughout the manuscript and ensured that they are defined at first usage.

(10) *p. 7, l.6; Kreitz et al. suggested such a stochastic process in their paper.*

We have added a citation to Kreitz, Furley, Memmert, & Simons, 2015, in which a stochastic process for noticing in inattentive blindness is outlined.

(11) *Typo: p.10, l.22: should be “than” instead of “that”*

(12) *Typo: p.12, l.6: “that the these”*

We have corrected these typos.

(13) *I did not get the vertical colored lines in Figure 4. Are they correct?*

We have added additional description to the caption in Figure 4 to clarify the meaning of those lines. They represent the edges of the invisible occluders that the unexpected objects emerge from, and are color-coded according to the direction the unexpected object travels when it enters the display.

(14) *I do not get the term "floating" in contrast to "edge onset". Perhaps there is a more intuitive term?*

We have changed this term to the hopefully more intuitive "mid-display onset," as Reviewer 2 raised a similar point.

(15) *P. 20/21: I do not fully get the paragraph that goes from p.20 to 21; rephrase. Perhaps some amount of this variability is due to random attentional shifts?*

We have rewritten this paragraph for clarity.

(16) *P. 21, l 9: make clearer that you mean the onset when speaking about the "narrow window of time"*

We have clarified this point.

(17) *P.21: what do you mean by "trajectories that began and ended near the middle of the display"? Is that correct?*

We have reworded this slightly; it refers to the same type of onset/offset behavior used in Experiment 1, in which the object emerges from and disappears behind invisible occluders that are positioned in the display and not at the edges.

Reviewer: 2

Comments to the Author(s)

Summary:

When people notice an unexpected event or object in an inattention blindness paradigm, do they notice immediately, or do they need sufficient time to notice and for the representation of the event or object to form? In three experiments, the authors convincingly show that "noticers" notice unexpected objects right away, and extra exposure time only contributing a small effect. When the object first appears close to the edge of the screen (where participants are attending for their primary task), people are more likely to notice it compared to when it appears closer to fixation. Finally, when people notice the objects, they immediately or very quickly represent it fully, as demonstrated by highly precise color recall.

Comments:

This paper is exceedingly clear: the question is clear (and interesting!), the hypotheses are clear, the writing is great, and the data are convincing, and the interpretation is warranted. I have no major comments, only a few optional suggestions or comments.

Minor comments:

Page 11, Table 1: Can you include the total number of people actually used in the experiments, to compare to the number excluded?

We have added two rows to this table that show the total number of subjects recruited as well as the total number retained in each experiment, in addition to the exclusions.

Also, about exclusions: These exclusions are very conservative, which ensures high quality of the results, but I'm wondering about the vision correction question. Do you have any sense of what correction they aren't using (and whether they'd actually need it for sitting at a computer)? I'm just surprised, it seems like a lot of people. Also, same for the technical issues: what problems did people have with it?

The vision correction question asks whether they “need glasses or contacts,” if they report that they “need glasses or contacts, but weren’t wearing them during the experiment,” they are excluded. This exclusion is conservative, as some people may only require correction for far distances but not require it for up-close tasks such as computer use. Since there can be so much variance both in monitor size and how close people elect to sit to their screens, we err on the side of excluding on this basis (and we have used this preregistered criterion in all of our past online studies of inattentional blindness).

For the technical issues, the most common reason for exclusion is reported display lags; this can happen if many tabs or programs are open at once. Some people also report, in “other issues,” that they realize they misunderstood the instructions, or that they found the task too difficult. We exclude on these ground in part to prevent the possibility that the display freezes or stutters while the unexpected object is onscreen, which could affect detection independently of any experimental manipulation.

We have added a bit of additional text to the manuscript to be more specific on these points.

Page 17, line 11: The "floating" onset is what was used in Experiment 1, correct? Might good to specify that (e.g. "(like in Experiment 1)"), so readers know it's not a new manipulation.

This is a great suggestion. We have changed this terminology to “mid-display onset” and explain it more carefully in the methods of Experiment 2, as well as highlight the parallels to the methods in the other experiments.

Page 20, Line 28: If the participants had been monitoring mid-line crossings, which is another standard inattentional blindness primary task, this would allow you to determine if the edge-advantage was due to attention, or something particular about edges per se.

This is a great point. The most systematic way to study the role of the locus of attention would be to move not just the onset point, as we did in Experiment 2, but also to move the locus of attention to different points in the display. We may explore those options in a follow-up series of experiments.

Reviewer: 3

Comments to the Author(s)

I read this this paper with great interest. Indeed, if the research question can seem "simple", it is very interesting and can provide a real contribution to the field.

In a general fashion, I have a very positive opinion about this paper. I really appreciated that it complies with new research practices and open science: All studies are pre-registered, high-powered, and materials, data and R scripts are available on osf. Note that the link to the pre-registration document is broken, so I was unable to check whether or not the performed analyses are the same as the pre-registered ones, etc (but I'm pretty confident they are).

We apologize for the error; the preregistrations were mistakenly left private at the time of submission. These have been made public, and the links have been verified.

Below are questions or minor concerns I had while reviewing the paper.

Why was the offset fixed at 2 s. before the end of the trial?

This choice was essentially arbitrary, beyond the need to have all of the unexpected objects exit the display with the same amount of time remaining to ensure that the retention interval was the same in all conditions. We do not expect the timing of the end of the trial to have an impact on the results.

More generally, I think that the authors could justify more the time frames they used. p. 11, why 2,7 s. for short exposure and 5 s. for long exposure? For instance, I think that the authors could justify the 5s exposure time in a better way than "a 5s exposure is typical for this sort of sustained inattention blindness task". I also wondered why the authors chose to use 2 exposure times and not 3? (even if I found some answer to this question in exp. 2).

We have more fully fleshed out the decisions behind our parameter choices in the general methods and the method sections for each experiment. For Experiment 1, we elected to use two exposure times that were sufficiently different in duration that they would be able to reveal whether noticing increases with time or stays constant, as we were also using the experiment to explore whether the self-reported location data seemed to provide a reliable signal. Since noticing appeared relatively unaffected by exposure time in that experiment, we added an additional, shorter exposure in Experiment 2 to further explore this relationship.

p 16 " in general, though, subjects who noticed the unexpected object positioned it close to its actual onset location with some precision ". Wouldn't it be possible to estimate more precisely the accuracy of the detection by calculating an error score for example?

The metrics we calculate, such as the proportion of localization reports appearing on the onset side of fixation and the average Euclidean distance of the reports to reference points, were intended to serve as an additional indication of localization. We hesitate to use the term "accuracy," since wherever subjects happened to perceive the object is a valid self-report.

However, we did want to characterize these reports in a quantitative way, which is why we report these metrics and present the Euclidean distances in tables.

In exp 3, I think that the conclusions of the study about the quality of the representation should more take in account that the unexpected object is very simple and has few features (a basic shape and one color). Conclusions about exposure times may be different with a more complex object.

We have added discussion of this point to both the discussion section of Experiment 3 and the Constraints on Generality section. It is true that subjects were already extremely accurate at reporting the color, so there simply may not have been room to observe any improvement to object encoding due to near-ceiling performance and that a more complex object would have yielded a different pattern of results.

p.27. The authors propose that "the onset event itself might trigger noticing". What if something in the unexpected object could grab attention (blinking, changing color)?

This is an interesting avenue for future research. It could be that the object changing its color or luminance serves the same role as an onset event and provides another opportunity to notice the object. Alternatively, it could be only the appearance of something new that triggers noticing, and anything that occurs afterwards has no impact on noticing. We are currently examining these possibilities in another series of experiments.

Finally, was there a deterioration in the primary task (i.e., counting the number of bounces) when participants detected the unexpected event?

There was no appreciable difference in counting errors. Noticers tended to have error rates about 2-3 percentage points higher on their final trial, corresponding to miscounting by about 1 bounce more than non-noticers. This is consistent with other inattentional blindness studies, which generally find little relationship between noticing and counting error rates (e.g. Beanland & Pammer, 2011).